# Female Genital Fibroblasts Diminish the In Vitro Efficacy of PrEP against HIV

**DOI:** 10.3390/v14081723

**Published:** 2022-08-04

**Authors:** Ashley F. George, Matthew McGregor, David Gingrich, Jason Neidleman, Rebecca S. Marquez, Kyrlia C. Young, Kaavya L. Thanigaivelan, Warner C. Greene, Phyllis C. Tien, Amelia N. Deitchman, Trimble L. Spitzer, Nadia R. Roan

**Affiliations:** 1Gladstone Institute of Virology, University of California at San Francisco, San Francisco, CA 94158, USA; 2Department of Urology, University of California at San Francisco, San Francisco, CA 94143, USA; 3Drug Research Unit, Department of Clinical Pharmacy, School of Pharmacy, University of California at San Francisco, San Francisco, CA 94143, USA; 4Women’s Health Clinic, Naval Medical Center, Portsmouth, VA 23708, USA; 5Departments of Medicine and Microbiology and Immunology, University of California at San Francisco, San Francisco, CA 94143, USA; 6Departments of Medicine and Veterans Affairs, University of California at San Francisco, San Francisco, CA 94143, USA; 7Lieutenant Colonel, United States Air Force, Medical Center, Women’s Health Clinic, Naval Medical Center, Portsmouth, VA 23708, USA

**Keywords:** HIV transmission, female reproductive tract, pre-exposure prophylaxis, fibroblasts

## Abstract

The efficacy of HIV pre-exposure prophylaxis (PrEP) is high in men who have sex with men, but much more variable in women, in a manner largely attributed to low adherence. This reduced efficacy, however, could also reflect biological factors. Transmission to women is typically via the female reproductive tract (FRT), and vaginal dysbiosis, genital inflammation, and other factors specific to the FRT mucosa can all increase transmission risk. We have demonstrated that mucosal fibroblasts from the lower and upper FRT can markedly enhance HIV infection of CD4+ T cells. Given the current testing of tenofovir disoproxil fumarate, cabotegravir, and dapivirine regimens as candidate PrEP agents for women, we set out to determine using in vitro assays whether endometrial stromal fibroblasts (eSF) isolated from the FRT can affect the anti-HIV activity of these PrEP drugs. We found that PrEP drugs exhibit significantly reduced antiviral efficacy in the presence of eSFs, not because of decreased PrEP drug availability, but rather of eSF-mediated enhancement of HIV infection. These findings suggest that drug combinations that target both the virus and infection-promoting factors in the FRT—such as mucosal fibroblasts—may be more effective than PrEP alone at preventing sexual transmission of HIV to women.

## 1. Introduction

Sexual intercourse is the most common means of HIV transmission, and worldwide, young women account for the majority of new cases of infections [1]. Pre-exposure prophylaxis (PrEP) has proven very efficient at curbing HIV transmission both in men and women [2,3,4]. However, women must adhere to a very stringent dosing schedule—6 days a week—to maintain optimal protection with oral tenofovir, whereas men who have sex with men (MSM) can achieve good protection with only 2 doses a week [5]. This disparity stems in part from the fact that the drug achieves higher concentrations in the gut or rectal mucosa than in the female reproductive tract (FRT) [5,6,7]. Strict adherence is also required for the dapivirine vaginal ring to be effective in preventing HIV transmission, in that the ring must not be removed during the month-long treatment [8,9,10]. While long-acting cabotegravir is given as an intramuscular injection every 8 weeks, adherence to the injection dosing schedule is also required for the regimen to be effective. As high adherence to PrEP is not always possible or realistic for at-risk women, lower PrEP concentrations are an unfortunate reality. Of note, in vivo tenofovir concentrations were below the limit of detection in the FRT of most women taking oral tenofovir [11,12], suggesting that there essentially is no lower limit when it comes to what levels of drugs are physiologically relevant. Under such sub-optimal drug concentrations, biological factors may further decrease the ability of PrEP to protect women against HIV. Indeed, factors such as vaginal microbiome composition [13,14,15], inflammation [16,17,18], and semen components [19,20,21,22,23] have been reported to increase HIV infection risk. 

For infection to be established in the FRT, HIV must breach the FRT epithelium to access the underlying vascularized stroma, where it infects resident immune cells [24] before propagating to the rest of the body. In contrast to the multi-layered epithelium of the vagina, the monolayer columnar epithelium of the endocervix and endometrium constitutes a weak barrier against pathogens transferred from the lower FRT through uterine peristalsis [25,26,27]. Recent reports suggest that the endometrium is a viable entry-point for HIV/SIV [27,28,29] and harbors HIV-permissive cells [30]. Our studies suggest that non-permissive endometrial stromal fibroblasts (eSFs) from the FRT mucosa can markedly enhance infection: CD4+ T cells can be infected at up to 100-fold higher rates when co-cultured with eSFs [31]. eSFs are therefore another factor that might increase women’s risk of infection under sub-optimal PrEP treatment adherence.

Assays testing the efficacy of PrEP should incorporate conditions that better mimic the FRT environment, allowing for a more accurate prediction of what is likely to occur during real-life transmission. Given the abundance of eSFs within the FRT [32,33], and their potent ability to enhance HIV infection, we set out to determine using in vitro assays whether these “infection-enhancing” cells may also affect the activity of PrEP drugs tenofovir disoproxil fumarate (TDF), cabotegravir, and dapivirine. We tested both high and sub-optimal concentrations of PrEP to reflect in vivo variation in adherence and waning concentrations after initial administration. We show that PrEP is significantly less effective in the presence of these genital fibroblasts, highlighting the possibility that dosage metrics established in vitro with T cells in isolation may not provide accurate estimates of activity in vivo.

## 2. Materials and Methods

### 2.1. Isolation and Culture of Endometrial Stromal Fibroblasts (eSFs)

Endometrial tissues were acquired from the Women’s Health Clinic of Naval Medical Center Portsmouth (NMCP) in Virginia, under established operating procedures. Subjects were pre-menopausal women between the ages of 18–49 years, confirmed to not be pregnant, and provided written, informed consent. The race of the subjects included those who were Caucasian (N = 1), African American (N = 1), or did not disclose (N = 3). Exclusion criteria included: (1) women who have been exposed to hormonal medications within 30 days of study enrollment, (2) women with any type of cancer and/or endometrial hyperplasia, (3) women with any immune-related comorbidities, and (4) women positive for COVID-19, or documented positivity for any sexually-transmitted diseases or infections. Endometrial tissue samples were obtained using a pipelle endometrial suction curette (Cooper Surgical, Trumbull, CT, USA). The eSFs were isolated and cultured as previously described [34]. Briefly, endometrial tissues were digested for 2 h with 6.4 mg/mL collagenase type I (Worthington Biochemical Corporation, Lakewood, NJ, USA) and 100 U/mL hyaluronidase (Sigma-Aldrich, St. Louis, MO, USA) in Hanks’ buffered salt solution with Ca^++^ and Mg^++^ (Life Technologies, Carlsbad, CA, USA). Digested material was run through a Falcon 70 μm cell strainer (Fisher Scientific, Waltham, MA, USA) to separate single cells and the eSFs were then further purified using selective attachment. The eSFs were cultured in serum-containing fibroblast growth medium (SCM; 75% phenol red-free Dulbecco’s modified Eagle’s medium (DMEM; Life Technologies) and 25% MCDB-105 (Sigma-Aldrich) supplemented with 10% charcoal-stripped fetal bovine serum (FBS; BenchMark from Gemini Bio Products, West Sacramento, CA, USA), 1 mM sodium pyruvate (Sigma-Aldrich) and 5 μg/mL insulin) until confluent. Fresh SCM was provided every 2–3 days during culture, and eSFs were used for experiments between passages 2–4. 

### 2.2. Isolation and Culture of Peripheral Blood Mononuclear Cells (PBMCs)

Primary peripheral blood mononuclear cells (PBMCs) were isolated from Trima reduction chamber buffy coats (Vitalant Blood Bank) by Ficoll-Hypaque density gradients and cultured in RPMI media supplemented with 10% FBS. To activate T cells, purified PBMCs were stimulated for 3 d with 10 ug/mL PHA (Sigma) and 100 IU/mL IL-2 (Life Technologies, Carlsbad, CA, USA). Activated PBMCs were then washed, passed through a 70 μm cell strainer (Fisher Scientific), and resuspended in media containing 20 IU/mL IL-2, for subsequent use in infection assays. 

### 2.3. Virus Preparation

The replication-competent CXCR4(X4)-tropic NL4-3.Luc [35] and CCR5(R5)-tropic transmitted/founder (T/F) F4.HSA [36] HIV-1 DNA constructs have been described previously. Briefly, NL4-3.Luc and F4.HSA HIV viral stocks were produced by PEI-mediated transfection (Thermofisher Scientific, Waltham, MA, USA) of 293T cells with proviral DNA expression plasmids. At 48 h post-transfection, 293T supernatants containing virus were filtered through a 0.22 um filter and concentrated by ultracentrifugation at 20,000 rpm (Beckman Coulter Optima XE-90) for 2 h at 4 °C. Viral titers were measured by the Lenti-X™ p24Gag Rapid Titer Kit (Clontech Laboratories Inc., Takara Bio Inc., Kusatsu, Shiga, Japan). For infection assays, viruses were used at final concentrations of 40–55 ng/mL p24Gag and the same batches of virus were used. 

### 2.4. Fibroblast Co-Culture Infection Assays

Fibroblast co-culture infection assays were conducted as previously described [31,37]. Briefly, PHA/IL-2 activated PBMCs (10^5^/well; N = 4 donors) were added to 96-well plates with confluent eSFs (N = 4–5 donors) or media alone. The indicated concentrations of TDF, cabotegravir, or dapivirine (Selleck Chemicals LLC, Houston, TX, USA) were incubated with cells for 30 min. The same PrEP drug stock was used for all infection assays and dilutions were prepared freshly prior to each assay. The indicated HIV reporter virus (NL4-3.Luc or F4.HSA) was added to PBMCs in the absence or presence of eSFs treated with different dilutions of PrEP drug and cultured for 3 d. Of note, our prior work demonstrated that similar HIV infection rates are observed when eSFs and PBMCs are autologous or from different donors, and that similar effects are observed with isolated CD4+ T cells and with PBMCs containing mononuclear cells [31]. For cells inoculated with NL4-3.Luc, PBMCs were harvested and processed with the Luciferase Assay System (Promega, Madison, WI, USA) and luminescence was monitored using an Enspire^®^ Plate Reader. To assess drug toxicity, a second set of replicate samples, grown in the presence and absence of HIV virus and PrEP drugs for 3 d, were processed with the CellTiter-Glo^®^ Luminescent Cell Viability Assay (Promega). Cell viability was monitored by luminescence using an Enspire^®^ Plate Reader. The limit of detection for luminescence ranged from 1 × 10^2^ to 1 × 10^10^ Relative Light Units (RLUs). For cells inoculated with F4.HSA, harvested PBMCs were assessed for infection levels by flow cytometry. Sextuplet experimental replicates were used for all infection assays.

### 2.5. Flow Cytometry

Cells were washed in FACS buffer (PBS + 2% FBS + 2 nM EDTA) and then stained for 30 min on ice with a cocktail containing fluorophore-conjugated antibodies diluted in FACS buffer. The antibody cocktail consisted of: APC/Cy7-CD3 (SK7, Biolegend, San Diego, CA, USA), PE/Cy7-CD4 (A161A1, Biolegend), APC-CD8 (SK1, Biolegend), FITC-CD24 (HSA, M1/69, BD Biosciences), and the LIVE/DEAD Zombie Aqua Fixable Viability Kit (Biolegend). Stained cells were washed twice with FACS buffer and then fixed with 2% paraformaldehyde for 20 min. UltraComp eBeads Compensation Beads (ThermoFisher) were used for each antibody individually for use as compensation controls, per the manufacturer’s protocol. Flow cytometric samples were run on a LSRFortessa (BD Biosciences, San Jose, CA, USA) and infection rates in CD4 + T cells were quantified using FlowJo software (FlowJo LLC, BD, Ashland, OR, USA). Productively-infected CD4+ T cells were identified as live, CD3+CD8- cells expressing the reporter gene HSA and that have downregulated cell-surface CD4. 

### 2.6. Quantitation of Intracellular Tenofovir-Diphosphate (TVF-dp) by Mass Spectometry

A sensitive liquid chromatography with tandem mass spectrometry (LC-MS-MS) method was developed and validated to measure tenofovir-diphosphate (TFV-dp) amounts in PBMCs and eSFs. TVF-dp and an internal standard (IS) were extracted from PBMC or eSF by protein precipitation using methanol (MeOH), followed by centrifugation and removal of the supernatant for analysis. Extracted samples were then injected onto an ultra-high performance liquid chromatography (UPLC) instrument, and resolved using an Thermo Hypercarb column with a gradient of ammonium bicarbonate buffer and acetonitrile mobile phases. A SCIEX 6500 triple quadrupole mass spectrometer in positive mode with electrospray ionization was used for detection. The method used a linear regression standard curve with a range from 0.5–200 ng/mL to estimate TFV-dp sample concentrations. TFV-dp sample concentrations were then converted to fmol/10^6^ cells. The lower limit of quantitation (LLOQ) for the assay was 0.5 ng/mL corresponding to 11 fmol.

## 3. Results

We tested the effect of eSFs on three inhibitors of HIV infection: the PrEP drugs TDF, cabotegravir, or dapivirine. TDF (formulated as an oral medication) and dapivirine (formulated as a vaginal ring) are inhibitors of viral reverse transcriptase, while cabotegravir (formulated as a long-acting injectable) is an integrase inhibitor. TDF and cabotegravir are approved by the U.S. Food and Drug Administration for PrEP against HIV in at-risk individuals, while dapivirine is approved by several African countries and is currently under review by others. PHA-activated PBMCs were exposed to NL4-3.Luc (a replication-competent luciferase reporter virus derived from NL4-3 [35]) in the absence or presence of eSFs and with increasing concentrations of the respective antiretroviral drug. Infection levels were then monitored 3 d later by luminescence.

We first established that treatment with PrEP drugs was not toxic to PBMCs at the tested drug concentrations, as shown by cell viability assays (Figure 1A–C). We then assessed HIV infection rates in PBMCs in the presence or absence of eSFs. In the absence of drugs, co-culture with eSFs enhanced infection rates ~7-fold (ranging from 4.7- to 9.37-fold; Figure 2A–C), consistent with prior reports [31,37]. TDF, cabotegravir, and dapivirine each inhibited HIV infection in a dose-dependent manner, both in the absence and presence of eSF. Importantly, however, infection rates were consistently higher in the presence of eSFs (Figure 2A–C).

As the initial experiments were performed with only one PBMC and one eSF donor, we next assessed the infection of PBMCs from one donor in the presence of eSFs from multiple donors (N = 4–5 donors). At lower drug concentrations (<3.125–4 nM), addition of eSFs increased infection rates on average 9.60-fold higher (range 6.94–15.88) for TDF, 3.40-fold higher (range 2.3–4.81) for cabotegravir, and 6.06-fold higher (range 2.54–18.44) for dapivirine (Figure 3A–C). At higher drug concentrations (range of 12.5–50 nM for TDF, 3.125–12.5 nM for cabotegravir, and 4–10 nM for dapivirine), HIV failed to infect PBMCs, but addition of eSFs restored infection to detectable levels. At even higher drug concentrations (200 nM for TDF, 50 nM for cabotegravir, and 25 nM for dapivirine), infection was blocked in both PBMCs and PBMCs co-cultured with eSFs. When PBMCs from multiple donors (N = 4 donors) were tested against eSFs from a single donor (Figure 4A–C), we observed similar trends. The IC50 values (the concentrations of PrEP drug where HIV infection rates are reduced by half of the maximal value, as determined separately for each condition) however, were not significantly different in the absence vs. presence of eSF (data not shown). Taken together, these results demonstrate that through a wide range of concentrations, the anti-HIV activity of PrEP drugs is diminished in the presence of eSF, independent of the PBMC or eSF donor used.

The experiments reported so far used an X4-tropic luciferase HIV reporter virus, which allows for high-throughput analysis of a large number of PrEP drug dilutions. We previously found that eSF-mediated enhancement of HIV infection occurs with both X4-tropic HIV and R5-tropic HIV viruses, including transmitted/founder (T/F) viruses [31]. As mucosal transmission of HIV is mediated by R5-tropic T/F HIV, we next investigated to what extent eSFs affect the antiviral activity of PrEP drugs against the R5-tropic NL-HSA.6ATRi-C.109FPB4.ecto reporter virus (HIV-F4.HSA), which encodes the R5-tropic T/F subtype C envelope 109FPB4 [36]. PHA-activated PBMCs (N = 3 donors) were incubated for 30 min in the absence or presence of eSFs, and in the presence of each of the tested PrEP drugs. Cultures were then infected with HIV-F4.HSA, and CD4+ T cell infection rates were assessed 3 d later by flow cytometry. A sequential gating strategy was used to identify live, CD4+ T cells as viable, singlet, CD3+CD8- cells (Figure 5A), so as to capture infected (HSA+) T cells that had downregulated surface CD4 after HIV entry [36,38,39]. In both the absence and presence of eSFs, HIV infection was inhibited in a dose-dependent manner by all three drugs (Figure 5B–D). However, the percentages of productively-infected CD4+ T cells was always higher in the presence of eSF, across all tested PBMC donors. These data suggest that eSFs reduce the efficacy of PrEP drugs against both X4-tropic and R5-tropic HIV viruses.

To determine the potential causes for PrEP’s reduced efficacy in the presence of eSFs, we assessed the effect of eSFs on drug uptake. Due to technical limitations in developing assays to measure intracellular levels of cabotegravir and dapivirine at this time, we focused our efforts on TDF. We incubated PBMCs for 3 days in the presence or absence of eSFs, with or without increasing concentrations of TDF (3.125–800 nM), and measured the concentration of intracellular tenofovir diphosphate (TFV-dp; the active anabolite of TDF) in both cell types by mass spectrometry. eSFs co-cultured with PBMCs had undetectable to low levels of TFV-dp when treated with 50 or 200 nM of TDF (Table 1). Levels of TFV-dp in eSFs exposed to 800 nM TDF were consistently detectable in all donors at low levels (Table 1), but this concentration had fully blocked HIV infection in the presence and absence of eSF (Figure 2, Figure 3 and Figure 4). These data together suggest at TDF concentrations where infection rates are enhanced in the presence of eSF, we do not detect any uptake of the drug by these cells, and therefore that the diminished activity of TDF in the presence of eSF is likely not because these cells are serving as a “sink” for the drug.

We then assessed intracellular TFV-dp in the PBMCs, co-cultured or not with eSFs. TFV-dp levels were below the limit of assay quantitation for PBMCs treated with 3.125 or 12.5 nM TDF (data not shown). At higher concentrations (≥50 nM TDF), TFV-dp levels in PBMCs increased in a dose-dependent manner, both in the absence and presence of eSFs (Table 1). However, levels of TFV-dp in PBMCs treated with 50, 200, or 800 nM tenofovir were not statistically different (*p* > 0.05) in the absence vs. presence of eSFs (Table 1). Altogether, these results suggest that the increased infection rates in the presence of eSFs is likely not due to PBMCs’ diminished access to TDF, but rather to the ability of eSFs to increase HIV infection rates.

## 4. Discussion

In this study, we evaluated to what extent PrEP drugs are protective in the context of HIV-enhancing fibroblasts—the most abundant cell type in the FRT. To mimic real-world variation in adherence to PrEP regimens, we tested both high and suboptimal concentrations of three different PrEP drugs. We demonstrated that female genital fibroblasts diminish the in vitro efficacy of the PrEP drugs TDF, cabotegravir, and dapivirine against HIV. These findings support the notion that testing of candidate PrEP regimens for women should include biological components of the FRT to more accurately assess conditions under which transmission will be blocked.

The mechanism by which eSFs reduce PrEP efficacy does not seem to entail reducing drug availability to HIV target cells. As evidence, we found no difference in TFV-dp levels between isolated PBMCs vs. PBMCs co-cultured with eSFs. Instead, our results point to an effect of eSFs on infection itself. Prior work from our group demonstrated that mucosal fibroblasts enhance HIV infection through two distinct mechanisms: (1) by directly transferring HIV virions to CD4+ T cells through *trans*-infection and (2) by conditioning CD4+ T cells to render them more permissive to HIV infection, a process that can be further enhanced under pro-inflammatory genital conditions [31,37]. We hypothesize that some of these mechanisms may underlie the compromised antiviral activity of PrEP drugs in the presence of eSFs, but further research is needed to decipher the precise mechanism(s).

In addition to fibroblasts, other factors within the FRT have been documented to increase HIV transmission rates and may also impair the anti-HIV activity of PrEP drugs. For example, semen, a major conduit for HIV transmission, and infection-enhancing fibrils from semen, also reduce the effectiveness of tenofovir and other antiviral regimens in vitro [19,21,40]. Furthermore, vaginal tenofovir gel is less effective in preventing HIV acquisition in women with bacterial vaginosis, likely because the bacteria rapidly metabolize and deplete tenofovir in vitro [13]. Dapivirine is similarly degraded by vaginal bacteria from women with dysbiosis in vitro [14]. Genital inflammation also increases the risk of HIV transmission [16,17,18]. Female genital inflammation is associated with a 3-fold higher risk of HIV acquisition [17], and can result from multiple factors, including semen exposure [41,42,43,44,45], bacterial vaginosis [15], sexually transmitted infections [15,46,47], and topical microbicides [17]. Further research is needed to determine the relative impact of these conditions on the efficacy of PrEP drugs, and to what extent they synergize with one another.

In conclusion, we demonstrated diminished efficacy of PrEP drugs against HIV by genital fibroblasts. Our results suggest that future pre-clinical studies testing PrEP activity should incorporate mucosal fibroblasts for a potentially more accurate prediction of effective drug concentrations that will prevent HIV transmission in vivo. Future research should also identify small molecules that inhibit the HIV-enhancing activity of fibroblasts and other biological factors within the FRT. These inhibitors could ultimately be added to existing PrEP regimens for a “combination PrEP” approach that can afford more potent and consistent protection, including under instances of suboptimal adherence. Such a combination approach would afford women more leniency in PrEP adherence, ideally closer to the two treatments a week that appear sufficient in MSM with oral tenofovir [5,6,7].

## Figures and Tables

**Figure 1 viruses-14-01723-f001:**
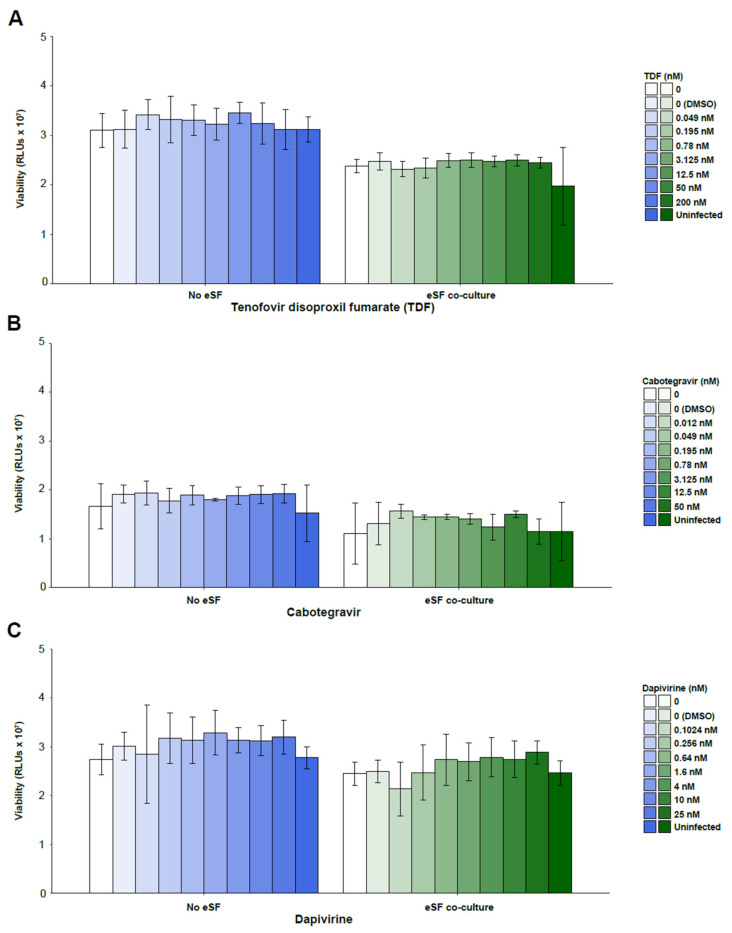
The tested concentrations of the PrEP drugs tenofovir disoproxil fumarate (TDF), cabotegravir, and dapivirine do not diminish PBMC viability in the absence or presence of eSFs. PHA/IL2-activated PBMCs were incubated with or without a HIV luciferase reporter virus (CXCR4-tropic NL4-3.Luc) and the indicated concentrations of (**A**) TDF, (**B**) cabotegravir, or (**C**) dapivirine in the absence or presence of eSFs. PBMC viability was monitored 3 days later by the CellTiter-Glo^®^ Luminescent Cell Viability Assay. Mean luminescence ± SD derived from sextuplet experimental replicates are shown (RLU/s, relative light units per second). The limit of detection for luminescence ranged from 1 × 10^2^ to 1 × 10^10^ RLUs. Non-significant (*p* < 0.05) in a group-wise comparison (one-way analysis of variance with a Tukey post-test).

**Figure 2 viruses-14-01723-f002:**
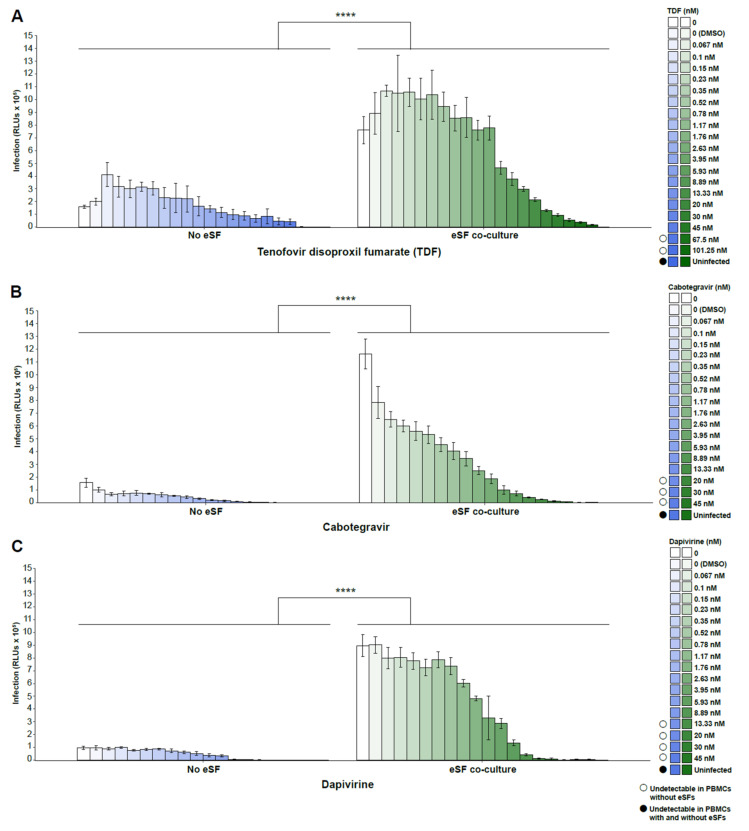
PrEP drugs tenofovir disoproxil fumarate (TDF), cabotegravir, and dapivirine lose antiviral activity in the presence of eSF. PHA/IL2-activated PBMCs were incubated with NL4-3.Luc at the indicated concentrations of (**A**) TDF, (**B**) cabotegravir, or (**C**) dapivirine in the absence or presence of eSF. Infection levels were monitored 3 days later by luminescence. Mean luciferase activities ± SD derived from sextuplet experimental replicates are shown (RLU/s, relative light units per second). The limit of detection for luminescence ranged from 1 × 10^2^ to 1 × 10^10^ RLUs. **** *p* < 0001 relative to no eSF coculture in a group-wise comparison (two-way analysis of variance with a Tukey post-test).

**Figure 3 viruses-14-01723-f003:**
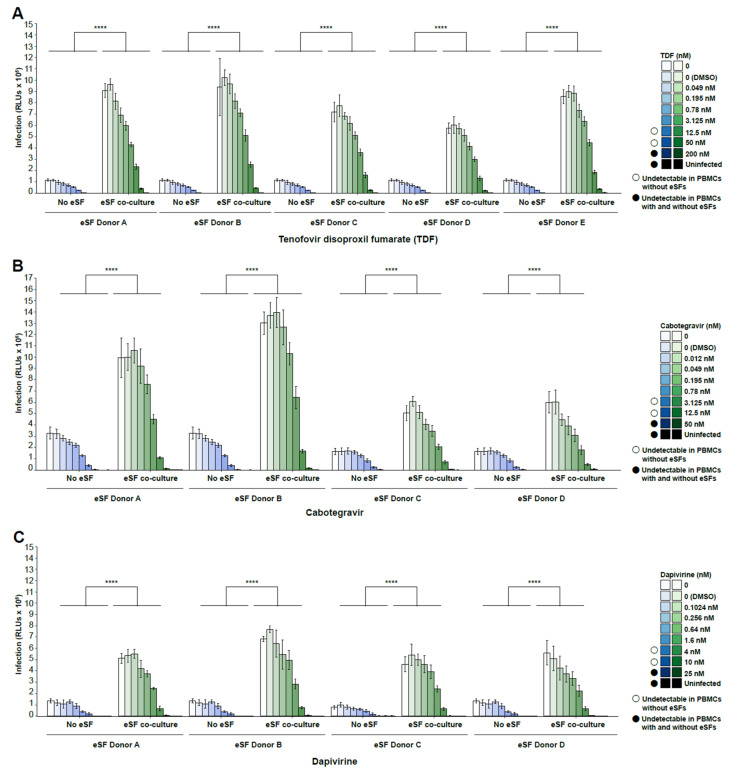
eSFs from multiple donors all diminish the in vitro efficacy of tenofovir disoproxil fumarate (TDF), cabotegravir, and dapivirine. Infection rates of PBMCs from a single donor exposed to (**A**) TDF, (**B**) cabotegravir, or (**C**) dapivirine in the presence or absence of eSFs from multiple donors. PHA/IL2-activated PBMCs treated with the indicated drug were mock-inoculated or inoculated with CXCR4-tropic NL4-3.Luc, in the absence or presence of eSFs (N = 4–5), and harvested 3 days later. Mean luciferase activities ± SD are shown. The limit of detection for luminescence ranged from 1 × 10^2^ to 1 × 10^10^ RLUs. Open circles indicate drug concentrations where infection was undetectable in PBMCs without eSFs. Filled circles indicate drug concentrations where infection was undetectable in PBMCs with or without eSFs. **** *p* < 0001 relative to no coculture in a group-wise comparison (two-way analysis of variance with a Tukey post-test).

**Figure 4 viruses-14-01723-f004:**
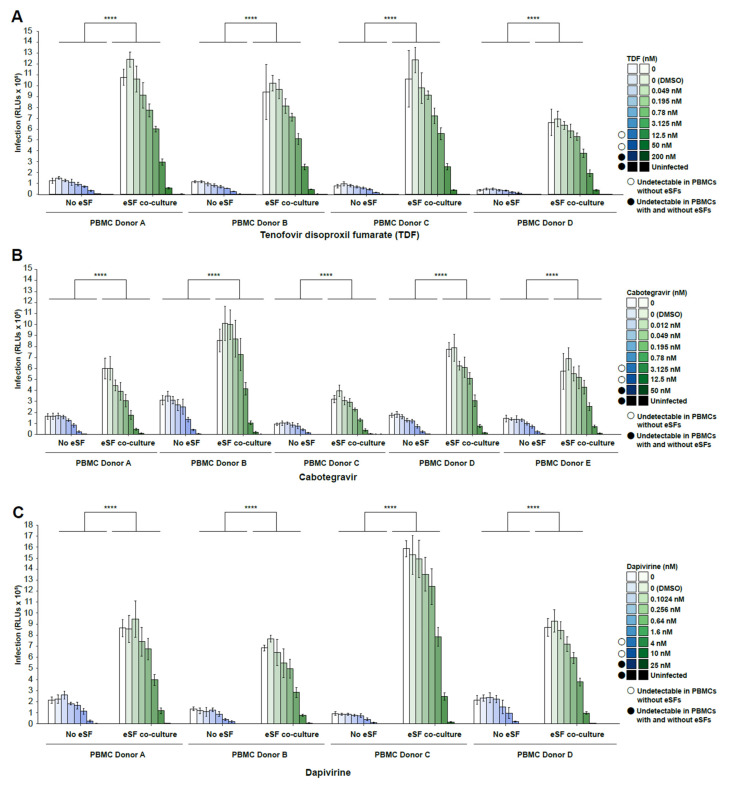
eSFs diminish the in vitro efficacy of tenofovir disoproxil fumarate (TDF), cabotegravir, and dapivirine in PBMCs from multiple donors. Infection rates of PBMCs (N = 4) exposed to increasing concentrations of (**A**) TDF, (**B**) cabotegravir, or (**C**) dapivirine are shown. PHA/IL2-activated PBMCs treated with the indicated drug were mock-treated or inoculated with NL4-3.Luc, in the absence or presence of eSF, and harvested 3 days later. Mean luciferase activities ± SD are shown. The limit of detection for luminescence ranged from 1 × 10^2^ to 1 × 10^10^ RLUs. Open circles indicate drug concentrations where infection was undetectable in PBMCs without eSFs. Filled circles indicate drug concentrations where infection was undetectable in PBMCs with or without eSFs. **** *p* < 0001 relative to no coculture in a group-wise comparison (two-way analysis of variance with a Tukey post-test).

**Figure 5 viruses-14-01723-f005:**
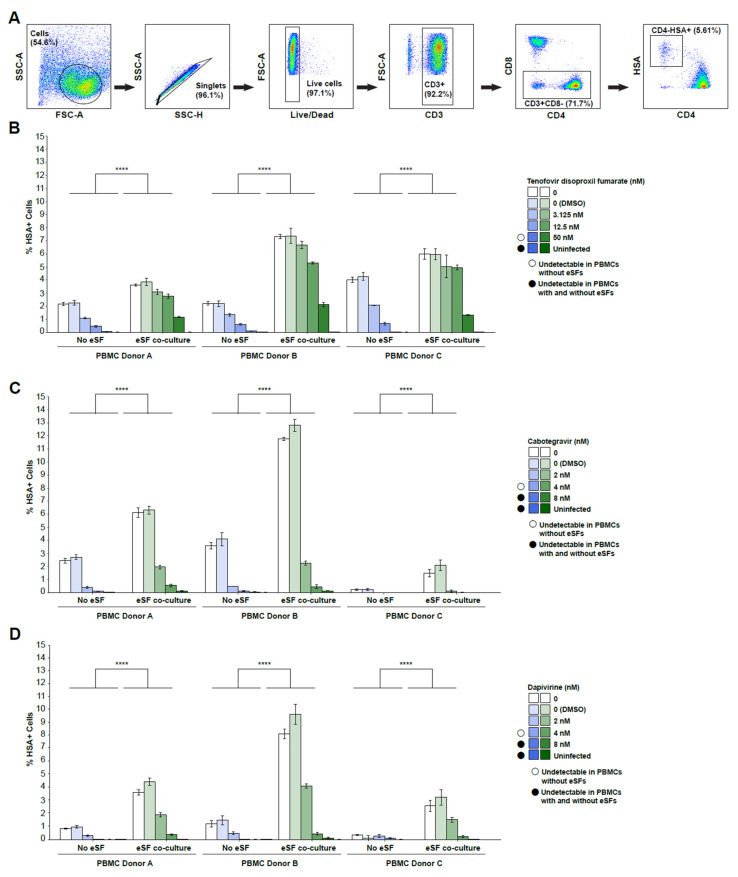
eSFs diminish in vitro efficacy of PrEP drugs against R5-tropic HIV infection. PHA/IL2-activated PBMCs exposed to increasing concentrations of tenofovir disoproxil fumarate (TDF), cabotegravir, or dapivirine cultured with or without eSF (N = 3), were infected with a CCR5-tropic transmitter/founder HSA-reporter virus, and monitored by flow cytometry for infection levels 3 days later. (**A**) Gating strategy for identification of infected cells. (**B**,**C**) eSFs diminish HIV infection rates in the presence of PrEP. Mean % of live, singlet CD3+CD8-HSA+ cells ± SD for (**B**) TDF, (**C**) cabotegravir, or (**D**) dapivirine are shown. Open circles indicate drug concentrations where infection was undetectable in PBMCs without eSFs. Filled circles indicate drug concentrations where infection was undetectable in PBMCs with or without eSFs. **** *p* < 0001 relative to no coculture in a group-wise comparison (two-way analysis of variance with a Tukey post-test).

**Table 1 viruses-14-01723-t001:** Tenofovir diphosphate (TFV-dp) concentrations in PBMCs cultured in the presence or absence of eSFs, with increasing concentrations of tenofovir disoproxil fumarate (TDF).

Sample	TDF Concentration	TVF-dp Concentration Median (IQR) ^a^
PBMCs	50 nM	0.79 ng/mL (0.62, 1.04)
	200 nM	3.50 ng/mL (1.81, 3.82)
	800 nM	18.00 ng/mL (11.46, 20.50)
PBMCs (co-cultured with eSFs)	50 nM	0.56 ng/mL (0.26, 1.30)
	200 nM	2.71 ng/mL (1.60, 6.04)
	800 nM	12.77 ng/mL (6.56, 29.73)
eSFs (co-cultured with PBMCs)	50 nM	0.53 ng/mL (<LLOQ, 0.58)
	200 nM	0.87 ng/mL (<LLOQ, 1.81)
	800 nM	3.98 ng/mL (0.79, 9.27)

LLOQ, lower limit of assay quantitation. Concentrations ≤ LLOQ are imputed as 0.5 ng/mL and no peak values are imputed as 0 ng/mL. ^a^ Pooled data from 3–4 eSF and 3–4 PBMC donors. Non-significant by paired *t*-test for PBMCs vs. PBMCs (co-cultured with eSFs) and PBMCs (co-cultured with eSFs) vs. eSFs (co-cultured with PBMCs).

## Data Availability

Not applicable.

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
