# Peer review of "Female Genital Fibroblasts Diminish the In Vitro Efficacy of PrEP against HIV"

_viruses, 2022, doi:10.3390/v14081723_

Round 1

Reviewer 1 Report

This manuscript by George et al. describes the effect of endothelial fibroblasts of HIV infection rates of PBMCs with three PrEP drugs, tenofovir, cabotegravir, and dapivirine. They sought to measure cell viability of PBMCs co-cultured with or without endothelial fibroblasts and they sought to understand a potential mechanism behind their finding of decreased efficacy of all three drugs in preventing HIV infection of PBMCs. This manuscript is a unique characterization of the interactions between PBMCs and mucosal fibroblasts in the female genital tract. This study provides data that identifies a potential mechanism for improving PrEP in women. This manuscript supports including biological factors from the female genital tract microenvironment in PrEP studies.

Introduction

Page 2 line 64: The authors state the eSFs are abundant within the FRT. The authors may include data that supports this statement, and their overall findings. What is the concentration of eSFs in the cervix compared to endometrial tissues?

Methods

There is limited discussion of the drugs used or the concentrations. Authors should include a justification for the concentrations used and discuss how these relate to clinically relevant concentrations in the discussion.

Results

Page 4 line 160: When discussing the first set of experiments, the authors do not state how many donors are used for either PBMCs or eSFs. While it is stated later in the results, it should be included earlier for clarity.

The authors chose to compare their data using a two-way analysis of variance. The limitation of this approach is that it primarily tests the contribution of SFs but does not inform on drug activity. Indeed, the figures and data suggest that the infectivity in the control groups is significantly different. An alternative approach would be to compare Ec50s and EC90s which would better inform on differences in drug potency in the presence of eSFs. Similarly, plotting Figure 2 (and Figure 3/4?) as dose response curves on the same x-axis would allow for visual comparison of efficacy in two groups and identify at what concentration of drug the presence of eSFs is no longer relevant for complete drug activity.

Reviewer 2 Report

·        The hypothesis of the article states: “Female genital fibroblasts diminish the in vitro efficacy of PrEP against HIV”. The authors provide an adequate proof of concept.

·        The hypothesis of the paper leads one to believe that only female mucosal fibroblasts play a role in diminishing the in vitro efficacy of all currently available PrEP agents.  The authors could reconsider reframing the hypothesis to include the mucosal pharmacology of PrEP agents that were investigated in the article.

·        The authors have not considered the other factors that could play an important role in diminishing the in vitro efficacy of PrEP. 

·        It does emphasise that it is in vitro and the focus on women

·        PrEP is a powerful tool that, as part of a comprehensive prevention package, has potential to significantly impact the HIV epidemic. PrEP effectiveness has always been believed to be dependent on the exposure and efficacy of antiretrovirals at the site of HIV transmission. However, of late, many researchers have postulated through in-vitro virological modelling and simulation trials (such as this in-vitro study) that host innate factors play an important role in efficacy.  

·        Understanding the role and mechanism of the complex mucosal micro-environment of the FRT plays in pharmacodynamic efficacy is one of the most important steps toward improving and optimizing PrEP regimens and dosing. Collectively, these in-vitro investigations are the first steps toward enabling better study designs and clinical trials, and adjusting therapeutic interventions for female patients.  

·        The authors and research team have provided enough laboratory and bio-informatic evidence to make the topic valuable in terms of future trials to assess PrEP efficacy.

·        Importantly it dispels the myth that the differences of efficacy observed to date of PrEP trials in women is attributable to poor adherence and strongly makes the case for looking at both biology and behaviour.

·     It is notable that while the tenofovir gel research and subsequent secondary analysis of biological factors contributing to enhancing understanding of why we need to look more closely at factors in the female genital tract is referenced the primary tenofovir gel paper published in Science in 2010 is not referenced, Abdool Karim, Abdool Karim et al. This trial finding catalysed the importance of understanding the role of biology and several of the arguments used in the discussion on potential mechanisms.

·    The literature review is lacking references that reflect the dynamic role of the gynecological microbiota, possible differentiation between the changes that could be related to upper and lower FRT and how these innate cells maintain homeostasis and function and notably differences by ethnicity and race.

·    The focus of this manuscript is not novel as several authors have been investigating gender differences in PrEP trial outcomes. The author’s have been meticulous in providing a thorough pharmacodynamic insight of the effects of endometrial stromal fibroblasts and peripheral blood mononuclear cells relationships with several PrEP agents – most of the previous work have focused only on tenofovir gel – the authors here have included oral PrEP, monthly dapivirine rings and 2 monthly CAB-LA making this manuscript more contemporary in terms of the expanded range of efficacious PrEP agents both in terms of new drugs and vehicles of delivery. This study provides a good framework and foundation for further in-vitro studies to continue research and for expanding and validating these findings to in vivo early clinical studies of new PrEP products being developed and evaluated.

·    Cell and virological culture is a difficult process, and as such, the authors should be commended for setting up this system.

·    Given the ethnic and racial diversity of the microbiome of the female genital tract the authors should consider validating their in vivo platform based on specimens from the USA for generalisability particularly in African women where the need for PrEP is highest   

·    The bioinformatic data provided by the study is valuable in terms of dosage and possible innate factors that need to be overcome for PrEP to be successful.  This highlights the need for a precision medicine protocol that can be designed and refined over this study's framework.

·    From a virological laboratory perspective, the study was well designed and thought through. The team utilized multiple methods in a stepwise fashion, that for the most part can be replicated with similar laboratory expertise and diagnostic equipment.

·    However, the article fails to make mention of any contaminating and/or confounding factors that would frequently occur in such an intricate study as this:  

·    These would include, but not limited to:

o   any confounding laboratory-based factors (sample pre-processing, sample processing, post-sample processing / input and output parameters), equipment, human or software error

o   Source of tissue and generalisability of findings given substantial ethnic and racial biases in microbiome composition

o   and how these issues were resolved to maintain the robustness of the findings

·      ·     In terms of the tissue that was harvested from donors, the authors did not explicitly qualify the donors in terms of:

·     Any other relevant clinical information available about the donors, eg. Co-morbidities, chronic illnesses, or even past COVID-19 infection.

·   HIV status,

·   Current or previous STI or vaginal inflammation ,

·   any medications that the patient may have been using recently, antibiotics, antifungals etc., that could alter the microbiota of the FRT

·    The ratio of pre-menopausal, menopausal, post-menopausal donors

·    and what procedures that patients were having done

·    From a laboratory perspective, the authors did not make mention if the study methods were designed around previous studies of a similar nature or if the study design was original.

·       ·        As mentioned above, the authors have provided valuable bioinformatic information but have not made mention of how the study design and execution took into account the possibilities of systematic and random errors generated outside and within the laboratory.  

·       ·        The overall conclusions that are specific to this study support the data that was produced and subsequently interpreted by the authors and research team.  However, the authors should round off the conclusion by offering an explanation and the possibilities of what these early findings could mean for PrEP dosages, the possibilities for patients and also for clinicians, and the potential uses of incorporating this study platform. These are in vitro observations and need to be validated through early clinical studies and underscore the importance of paying attention to FGT biology and its role in reducing potency particularly as new generation products are being developed to address issues of adherence.  Support the authors recommendation that innovation of PrEP must combine both behavioural and biological considerations.    

·        ·        The summary is concise and presents the author’s research question and the reasons why there is a need for such a study.  The authors have also summarised the bioinformatic data well enough so that the reader perusing pertinent information will gain an immediate grasp of the overall contents.

      ·        On first read the first sentence does articulate a common fallacy to explain the gender differences in PrEP efficacy to behaviour ie adherence. The second sentence poses the role of biology as a potential additional explanation. However there is a compelling body of evidence on biological factors both viral and immune contributing to these gender differences that the authors cite (Klatt, McKinnon, Ngcapu, Passmore etc as examples).  The somewhat novel facet is specifically focusing on fibroblasts.

Reviewer 3 Report

George et al. evaluated the effect of human endometrial stromal fibroblasts (eSFs) on HIV-1 infection of PHA-stimulated PBMCs in the presence and absence of multiple concentrations of the antiretroviral inhibitors tenofovir disoproxil fumarate (TDF), dapivirine (DAP), or cabotegravir (CAB). Both a lab-adapted reporter virus and a clinical isolate with a reporter gene were evaluated with PBMC from multiple donors. In all cases co-culture of PBMCs with eSFs lead to increased infection both in the presence and absence of TDF, DAP, or CAB.

In addition, the authors also looked at the effect of eSF co-culture on active, diphosphorylated tenofovir, TFV-dp. This metabolite was not detected at some concentrations with clear antiretroviral activity. When it was detected, it did not appear that significantly different concentrations were noted.

Overall, it is clear that eSFs can enhance HIV-1 infection even in the presence of different antiretroviral drugs used for PrEP, which the authors contend could contribute to less efficacy in women. However, there is still no clear mechanism and no discussion was provided on SFs in the colon and how this can influence HIV-1 transmission in men who have sex with men (MSM).

Major comments:

1. Lines 37-39: This statement requires a citation. It likely comes from Cottrell et al. (reference 5), who modeled this.

2. The legend for Figure 1 indicates that PBMC viability was measured by infection with the luciferase reporter virus. Presumably this is incorrect, as the methods section indicates the CellTiter Glo assay was used. Assuming these data show a different type of viability measurement (e.g. the figure legend is incorrect), it appears that co-culture of PBMCs with eSFs decreases viability, particularly in Figure 1A. Was that statistically significant?

3. Does it matter that the eSF donors and the PBMC donors were apparently different, which could the case if infected cells were present from a sexual partner? Are similar effects observed when the donor of both cell types are the same?  Does the presence of other mononuclear cells (i.e. CD8+ cells, B cells, monocytes) in the culture effect the outcome?

4. The TFV-dp data are unclear. First, Table 1 indicates that the results are pooled for 3 eSF donors, yet no standard deviations are noted. In addition, it is concerning that an unpaired t test was performed in some cases (PBMCs vs. PBMCs+eSFs) but not in others (PBMCs vs. eSFs in co-cultures). What is the rationale for this? It is doubtful that a 2-fold difference in TDF-dp concentrations accounts for the difference in infectivity even with enhanced virus replication, as the drug is in vast excess.

It is also surprising that 3-12.5 nM TDF, which inhibits nearly all HIV-1NL4-3 infection, does not lead to any detectable active metabolite in PBMCs.

5. The authors conclude that the presence of eSFs in the female genital tract may, in part, contribute to reduced efficacy of PrEP in women based on the results presented in the manuscript. However, the colon also contains stromal fibroblasts. Have these cells been tested for the same effect?  Discussion of the role of these cells in MSM transmission should be provided.

Minor comments:

1. Line 35: UNAIDS citation should be included in References.

2. Line 150: tenofovir disoproxil fumarate is commonly abbreviated as TDF and it is a prodrug of tenofovir. Technically, it is not tenofovir as indicated here.

3. Lines 100 and 156: the authors should clarify whether the NL4-3.Luc is a replication-competent (i.e. env+) or replication-defective (i.e. env-, requires pseudotyping) molecular clone. Reference 27 describes both. If the latter, it is unclear how pseudotyping was performed.
